# Fibrinolysis in Dogs with Intracavitary Effusion: A Review

**DOI:** 10.3390/ani12192487

**Published:** 2022-09-20

**Authors:** Andrea Zoia, Michele Drigo, Marco Caldin, Paolo Simioni, Christine J. Piek

**Affiliations:** 1Division of Internal Medicine, San Marco Veterinary Clinic, Viale dell’Industria 3, 35030 Veggiano, Italy; 2Department of Medicina Animale, Produzione e Salute, Padua University, Viale dell’Università 16, 35020 Legnaro, Italy; 3Laboratorio d’Analisi Veterinarie San Marco, Viale dell’Industria 3, Veggiano, 35030 Padua, Italy; 4Department of Cardiologic, Thoracic and Vascular Sciences, University of Padua Medical School, Via Giustiniani 2, 35128 Padua, Italy; 5Department of Clinical Sciences of Companion Animals, Faculty of Veterinary Medicine, Utrecht University, 8 Heidelberglaan, 3584 CS Utrecht, The Netherlands

**Keywords:** ascites, pleural effusion, FDPs, D-dimer, ROTEM, hyperfibrinolysis, primary hyperfibrinolysis

## Abstract

**Simple Summary:**

In blood vessels there is a balance between clot formation and its dissolution. Fibrinolysis normally allows the breakdown of blood clots during the healing of injured blood vessels. This process is mediated by the activation of a blood enzyme (plasmin) which breaks down a meshed protein (fibrin) which holds blood clots at the site of the vessel injury. In some diseases, the activation of plasmin becomes excessive, leading to bleeding tendencies (hyperfibrinolysis). Under normal conditions, abdominal and thoracic cavities are filled with a small amount of fluid deriving from the blood. The results of recent studies have shown that, in dogs, all types of pathologic intracavitary fluids have an increased fibrinolytic activity. This increased fibrinolytic activity is also present in their blood, in some cases reaching a hyperfibrinolytic state. Hyperfibrinolysis and bleeding tendencies have also been documented in cardiopathic dogs with ascites. The latter result is surprising considering that thrombotic events are commonly documented in humans and cats with some cardiac diseases.

**Abstract:**

Physiologic fibrinolysis is a localized process in which stable fibrin strands are broken down by plasmin in response to thrombosis. Plasmin activation can also take place separately from the coagulation process, resulting in pathologic fibrinolysis. When plasmin activation exceeds the neutralizing capacity of plasmin inhibitors, severe bleeding can potentially take place. Although the processes which regulate coagulation and fibrinolysis in the blood are well known, it is less clear as to what extent the same processes take place in the body cavities and whether they influence systemic hemostasis. The results of the studies herein cited demonstrate that coagulation followed by fibrinogenolytic/fibrinolytic activity takes place in all kinds of canine ascitic and pleural fluids. Moreover, systemic clotting abnormalities suggesting primary fibrinolysis/primary hyperfibrinolysis (i.e., elevated plasma fibrin/fibrinogen degradation products [FDPs] and normal D-dimer concentrations with fibrinogen concentrations ≤ 100 mg/dL or above this cut-off, respectively) occur in dogs with intracavitary effusion. Enhanced fibrinolytic activity in dogs with intracavitary effusion can also be detected using rotational thromboelastometry (ROTEM), although the degree of agreement between ROTEM and FDPs, D-dimer and fibrinogen concentrations is poor. Finally, contrary to the thrombotic events commonly documented in some humans and cats with cardiac diseases, bleeding tendencies due to primary fibrinolysis/primary hyperfibrinolysis have been documented in dogs with cardiogenic ascites.

## 1. Introduction

Fibrinolysis is the process in which stable fibrin strands are broken down by plasmin, [1] and plays an important role in regulating the size of the clot [2]. The regulation of fibrinolysis is obtained by means of a dynamic balance between pro-fibrinolytic and anti-fibrinolytic processes regulated by complex interactions between circulating proteins essential for fibrinolysis, clotting factors, and endothelial cells [3]. Although the processes which regulates coagulation and fibrinolysis in the blood are well known, it is less clear as to what extent the same processes take place in the body cavities and whether they influence systemic hemostasis.

The first aim of this review was to describe the pathophysiology, classification and detection of fibrinolysis in clinical practice. The second aim was to summarize the findings in the literature which refer to the intrinsic fibrinolytic activity of the intracavitary fluid in dogs and to describe their systemic fibrinolytic status.

## 2. Pathophysiology of Fibrinolysis

Coagulation and fibrinolysis are strictly controlled by the interaction of substrates, activators, inhibitors, cofactors and receptors [4]. The conversion of fibrinogen to fibrin by thrombin leads to the exposure of lysine binding sites in fibrin. This promotes the binding of the circulating tissue plasminogen activator (tPA) and plasminogen to the fibrin network [2]. Plasmin is the major fibrinolytic protease and derives, in the majority of cases, from the conversion of plasminogen to plasmin by the tPA [4]. By means of a positive feedback mechanism, the tPA is cleaved from a single chain to a more active two-chain polypeptide by plasmin. Therefore, fibrin regulates its own degradation by binding both tPA and plasminogen on its surface, localizing and increasing the generation of plasmin [4]. In fact, fibrin increases tPA catalytic efficiency for plasminogen activation to plasmin up to 500-fold [2]. In addition to the tPA, the urokinase plasminogen activator (uPA) can also activate plasminogen to plasmin. As compared to the tPA, the uPA has lower affinity for plasminogen, does not require fibrin as a cofactor, and, for the most part, appears to act in the extravascular setting [5].

The tissue plasminogen activator and plasminogen binding to fibrin lysine sites can be blocked by the thrombin-activatable fibrinolysis inhibitor (TAFI) or by therapeutic lysine analogues such as epsilon aminocaproic acid and tranexamic acid. When activated by thrombin, the TAFI down-regulates fibrinolysis by eliminating C-terminal lysine residues from partially degraded fibrin, thus decreasing plasmin generation, stabilizing fibrin thrombi, and establishing a regulatory connection between coagulation and fibrinolysis [2]. Once plasmin is formed, it cleaves fibrin, generating fibrin/fibrinogen degradation products (FDPs), including D-dimers [2]. Fibrin-associated plasmin and the tPA are protected from their major inhibitors, α2-plasmin inhibitor (α2-PI) and plasminogen activator inhibitor-1 (PAI-1), respectively [4]. On the contrary, free tPA in the blood can be inhibited by PAI-1, forming inactive tPA/PAI-1 complexes; free plasmin in the blood can rapidly be inhibited by α2-PI, forming inactive plasmin/α2-PI complexes [2], (Table 1 and Figure 1).

## 3. Classification of Fibrinolysis

Fibrinolysis primarily blocks the unnecessary buildup of intravascular fibrin [5,6]. It generally occurs in response to activation of the coagulation, and is closely related to it. The rapidity with which a clot is lysed depends on: (1) the rate of tPA secretion by the endothelium, (2) the velocity of the tPA inhibition by PAI-1, (3) the rate of tPA clearance by the liver, and (4) enhancement of the tPA activity by intravascular fibrin [2]. Based on the trigger for fibrinolysis, its grade of severity, and whether it is localized or systemic, several types of fibrinolysis have been described.

**Basal fibrinolysis** or **natural fibrinolysis**, the ongoing removal of fibrin deposits, ensures blood fluidity while impeding thrombosis [7,8]. In humans, this type of low-level systemic fibrinolysis has a circadian variation [9] and can be affected by exercise [10].**Physiologic fibrinolysis** is a localized type of fibrinolysis in response to clot formation at the sites of injury or inside blood vessels (thrombosis), and is vital for re-establishing blood flow [6]. It is mediated by fibrin-bound plasmin. This type of fibrinolysis is localized to the site of the clot or the thrombus.**Primary fibrino(geno)lysis** (PF) occurs independently of the intravascular activation of the coagulation process, with the generation of plasmin without concomitant thrombin generation [11], and lysis of preformed fibrin and fibrinogen [11,12,13]. Thus, it is regulated by plasmin free in the plasma, and is a generalized fibrino(geno)lysis. When the production of plasmin within the general circulation exceeds the neutralizing capacity of the plasmin inhibitors, which could eventually cause severe bleeding [6,12], the disorder is called **primary hyperfibrinolysis [6]**, **primary hyperfibrinogenolysis** (PHF) [6,12,14], or **pathologic fibrinolysis** [6]. It has recently been proposed that if the intensity of PF reduces plasma fibrinogen concentrations to ≤100 mg/dL, PHF could be present [15], since the risk of bleeding increases below this fibrinogen concentration [16].When fibrinolysis takes place as a correct response to persistent thrombin generation, it is called **reactive or secondary fibrinolysis** (SF). Secondary fibrinolysis is therefore a consequence of the activation of a coagulative process, often due to disseminated intravascular coagulation, which triggers the endothelium to produce an increased amount of tissue plasminogen activator, with activation of plasminogen to plasmin [11]. This consumptive coagulopathy can lead to a hyperfibrinolytic state with a deficiency in circulating fibrinogen and a tendency to increased bleeding [5]. Lysis of cross-linked fibrin during SF will generate both FDPs and D-dimers [12,13]. This phenomenon maintains blood vessels patent by the resolution of redundant clots [17]. Secondary fibrinolysis, in addition to its presence in virtually every patient with disseminated intravascular coagulation [6], has also been reported in inflammatory diseases, such as sepsis [6,18,19].Similar to the definition of PHF, if SF is so intense as to potentially cause severe bleeding, the disorder should be called **secondary hyperfibrinolysis** (SHF), and fibrinogen concentrations ≤ 100 mg/dL could represent a marker of such fibrinolytic intensity [16].

Finally, it is of interest to note that various other definitions of hyperfibrinolysis are present in the literature. Schöchl et al. [20], and other authors [21,22,23], have arbitrarily used the term ‘hyperfibrinolysis’ to describe lysis greater than a certain maximal amplitude on thromboelastography (TEG)/rotational thromboelastometry (ROTEM) testing [24]. However, there has been uncertainty when using this viscoelastometric-associated terminology, since hyperfibrinolysis customarily refers to a situation in which the fibrinolytic activity is greater than the fibrin formation, clot integrity is threatened, and there is clot breakdown [24], rather than a loose term used simply to describe the increased evidence of fibrinolysis [22,25,26,27,28]. Therefore, the term ‘TEG/ROTEM hyperfibrinolysis’ has been suggested in relation to the viscoelastometric measurements [24].

In PHF, the exaggerated production of plasmin within the general circulation exceeds the neutralizing capacity of the plasmin inhibitors [11]. Due to its nonspecific proteolytic activity, the excessive free plasmin concentration in PHF results in the following: generalized fibrinogenolysis, increased production of FDPs, degradation of coagulation factors V, VIII, IX and XI [12], and degradation of any pre-existing fibrin localized in hemostatic clots [11,12,13], which can eventually lead to severe bleeding [6]. Together with exaggerated serum plasmin concentrations, there are other enzymes, such as serum tryptase or non-plasmic polymorphonuclear elastase, which have been reported to be possible causes of PHF when their serum concentration exceeds the neutralizing capacity of the plasmin inhibitors [29,30]. In humans, this has been tied to acute conditions, such as shock [12], aortic surgery [31], liver transplantation [32], acute leukemia [33], and treatment with thrombolytic drugs. Primary hyperfibrinolysis can also occur with tumors [34], and in other chronic conditions, such as liver disease [35], or following peritoneovenous shunting [36,37,38,39]. In small animals, PF and/or PHF have been suggested to occur in dogs with intracavitary effusion, the focus of the second part of this review [15,40,41,42], and possibly in dogs with angiostrongylosis [43] or in greyhounds post-surgery [44]. In cats with cardiogenic pleural effusion, PF/PHF have been hypothesized to be a protective factor in aortic thromboembolism occurrence [45]. Finally, using ROTEM, hyperfibrinolysis has also been detected in cats with traumatic hemoabdomen or hemothorax [46], and in dogs with angiostrongylosis [46,47,48], neoplasia [46], liver disease [46,49,50], traumatic hemorrhagic shock [51], and other diseases [15], as well as in apparently healthy dogs [46].

## 4. Detection of Fibrinolysis in Clinical Practice

It is difficult to measure fibrinolysis directly. For in vitro assays, predicting the development of thrombosis due to hypofibrinolysis or bleeding due to hyperfibrinolysis is difficult due to the complex interplay of hemostatic and fibrinolytic proteins. Moreover, the interference of variables, such as hyperlipidemia and inflammatory mediators, has rendered the creation of a predictive fibrinolytic assay challenging [5]. For the aforementioned reasons, progress and standardization of fibrinolytic methods have developed more slowly than coagulation testing, and routine high-throughput screening tests for fibrinolysis are still lacking [52].

A variety of assays currently available for the clinical evaluation of the fibrinolytic system are listed below:**Euglobin clot lysis time** has been developed as a global test of plasma fibrinolysis. This assay is carried out on citrated plasma samples with the addition of calcium and phospholipid vesicles; the resulting fibrin clot is then lysed by the addition of exogenous tPA [5]. Euglobin clot lysis time is sensitive to the level of active tPA in the sample; however, the assay is too slow to evaluate patients with active bleeding, up to 4 h of run time in healthy humans. It is not available in the majority of laboratories, and standardized methodology is lacking [2]. Therefore, this test is not routinely utilized in veterinary medicine. In human patients with different diseases associated with venous and arterial thrombosis, hypofibrinolysis measured by this test can predict the risk of thrombosis. However, excessive fibrinolytic activity, as defined by clot lysis time, has not shown notable predictive power in bleeding disorders [5].**Viscoelastic methods** are one of the few ways available for investigating coagulation and fibrinolysis in citrated whole blood, therefore accounting for the contribution of platelets and other blood cells [52]. Three instruments are currently used in veterinary and human medicine to monitor viscoelastic changes in the blood: ROTEM, TEG, and Sonoclot analyzers. In the majority of published studies, ROTEM or TEG have been used [15,46,47,48,51,53,54,55]. All of the approaches detect variations in blood viscosity by the movement changes of a pin through anticoagulated whole blood in a cup after triggering coagulation by an activator [52]. For example, considering the ROTEM delta machine, differential testing on a patient is accomplished with a single unit, using four channels. The results are shown as a set of clotting and lysis curves called temograms. Up to 32 parameters are derived which describe clotting and lysis kinetics and clot properties. Numerous tests exist for TEG which are similar, but not identical to the ROTEM tests; however, there are differences with regard to measurement technique and reagents [52]. Concerns have been raised regarding the sensitivity of viscoelastic methods for the identification of fibrinolysis in all patients. In fact, humans having a low to moderate degree of ongoing fibrinolysis, with increased fibrinolytic markers, such as plasmin/α2-PI complex, D-dimer, and tPA, escaping identification by ROTEM [52]. More recently, to overcome this problem, tPA-modified TEG and ROTEM assays were developed and were found to be more sensitive in revealing fibrinolysis, both in humans and dogs [54,55,56,57,58,59,60].Assays for **D-dimer** and **FDPs** are both available in veterinary [61,62,63,64,65] and human medicine [52]. A D-dimer is formed secondary to the FXIII crosslinking of adjacent D-domains in fibrin polymers. Hence, circulating D-dimers is an indication of fibrin breakdown and ongoing SF. Therefore, SF is different from fibrinogenolysis [37] in which only FDPs, but not D-dimers, are increased. It is known that antibodies to ‘D-dimer’ may cross react with high-molecular-weight FDPs; the type of such FDP fragments changes according to the disease [52]. The clinical application of D-dimer testing is limited to excluding thrombosis, since D-dimer test methods tend to have good sensitivity, but poor specificity, frequently leading to false-positive results [52].

It is of interest to note that standard criteria for diagnosing PF/PHF are lacking in the published literature. In PF/PHF, the production of FDPs should be increased, but not the production of D-dimers [12]. Therefore, discordant results between elevated FDPs and normal D-dimers have been suggested to be a strong indicator of PF/PHF in humans, and they have been used to reach this diagnosis in a clinical setting [12,29,66].

4.**Other clinical assays** reported in the human literature are represented by the tPA antigen, tPA activity, the PAI-1 antigen, PAI-1 activity, and antiplasmin activity while **other research assays** described in the human literature are represented by the tPA/PAI-1 complex, plasminogen activity, the plasmin/α2-PI complex antigen, the uPA antigen, uPA activity, the TAFI antigen and TAFI activity [2].

## 5. Fibrinolytic Activity of the Intracavitary Fluid in Dogs

Ascites and pleural effusion are the abnormal accumulation of fluids in the peritoneal and pleural cavities. These intracavitary effusions can be classified according to the pathophysiology of their formation in exudate, transudate, and hemorrhagic and chylous effusions [67]. Exudates are considered to be effusions which are formed secondary to increased permeability of the peritoneal or pleural surfaces; however, transudates result from systemic conditions which alter Starling forces and are formed across normal peritoneal or pleural surfaces [67,68,69]. Finally, chylous effusions normally occur in the thorax and are the result of a thoracic duct leak of lymph rich in triglycerides. This lymph arises from the reabsorption of the interstitial fluid of the lower extremities, abdomen (including intestinal lacteals) and thorax, and, for the most part, resembles a plasma transudate. Therefore, except for hemorrhagic effusions wherein the abdominal or pleural fluids accumulate due to rupture of the blood vessels or secondary to clotting disorders, the remaining intracavitary effusions originate secondary to the exudation or ultrafiltration of the plasma. Nevertheless, regardless of the mechanism of formation, all the intracavitary fluids contain the proteins involved in coagulation [70,71] in a milieu in which the actions of these proteins are not well regulated, owing to the lack of other constituents of the hemostatic system (e.g., platelets, vascular endothelium).

Two recent studies have investigated whether coagulation and fibrinogenolytic/fibrinolytic activity occurred in intracavitary effusion by measuring the plasma and the effusion fibrinogen, and the FDPs and D-dimer concentrations of 70 dogs with ascites and 33 dogs with pleural effusion of different pathophysiological origins [41,72]. Abdominal and pleural effusion fibrinogen concentrations in these 103 dogs were undetectable in most cases and significantly lower than those in the plasma while the concentrations of FDPs and D-dimers in the abdominal and pleural fluids were significantly greater than those in the plasma [41,72].

The results of these two studies would therefore suggest that (1) in the abdominal and pleural fluids of dogs, coagulation activation and fibrinolysis existed in nearly all the dogs, and (2) that this phenomenon took place regardless of the underlying mechanism which led to the formation of the intracavitary effusion. The first statement was supported by the finding that fibrinogen concentrations in dogs were significantly lower in the intracavitary fluid as compared with the plasma, whereas the FDPs and D-dimer intracavitary fluid concentrations were significantly higher than those in the plasma. Taken together, these results indicated that fibrinogen, when entering into the abdominal or pleural cavities, was transformed into cross-linked fibrin, and was then lysed to form D-dimers. The increase in FDPs would support this finding; however, it also suggested that some fibrinogen, fibrin monomer or polymers may be lysed even before being transformed into cross-linked fibrin. The results of these two studies were in agreement with experimental results in dogs having pleural effusion [73,74], and also with what was found in 3 human clinical studies in which all the ascitic fluids evaluated were seen to have fibrinolytic activity, independent of the underlying mechanism of formation [70,75,76]. Finally, horses with different types of pathological abdominal effusion also have significantly higher ascitic D-dimer concentrations than plasma concentrations, suggesting that there is fibrinolytic activity within the effusion [77]. A possible biological explanation for these findings relies on the fact that fluids of virtual cavities, of which abdominal or pleural effusions are a pathological manifestation, must be clot free in order to allow the smooth sliding of organs over one another; therefore, clots must be rapidly lysed. This statement is also supported by the clinical observation that abdominal and pleural effusions rarely form clots in vivo and also by experiments in dogs showing that blood introduced into the pleural or abdominal cavities remains fluid owing to previous coagulation and defibrination [73,78].

## 6. Systemic Fibrinolysis in Dogs with Intracavitary Effusion

Abdominal and pleural fluids inherently have fibrinolytic activity [41,70,72,75] and are in continuous exchange with the systemic circulation [75,79]. Two studies regarding dogs with abdominal and pleural effusion of different pathophysiological origins evaluated whether clotting abnormalities, suggestive of PF/PHF, were more frequent in these dogs. Defining PF as discordant results between elevated plasma FDPs and normal D-dimer concentrations with fibrinogen concentrations > 100 mg/dL and PHF as discordant results between elevated plasma FDPs and normal D-dimer concentrations with fibrinogen concentrations ≤ 100 mg/dL [15], the combined data of these two studies showed that PF/PHF occurred more frequently in the 103 dogs with intracavitary effusion (70 dogs with ascites and 33 dogs with pleural effusion) in comparison to two age, sex and breed-matched control populations of clinically healthy and sick dogs without intracavitary effusion (Table 2) [41,42]. A similar result was found in a different study regarding 32 dogs with intracavitary effusion (25 with only abdominal effusion, six dogs with concomitant abdominal and pleural effusion, and one dog with only pleural effusion) [15]. In this group, the frequency of PF/PHF, assessed based on the above given definition, was significantly higher (63% vs. 31%, respectively) as compared to a control group of 32 dogs without intracavitary effusion, individually matched to the dogs studied based on type of underlying disease. Of the 20 dogs with intracavitary effusion, 15 had PF and the remaining five dogs had PHF [15]. Also in this case, none of the control dogs had PHF and only 10 of the control dogs were classified as having PF [15].

It is interesting to note that at the level of the systemic circulation, PF/PHF occurred in dogs with all the pathophysiological types of intracavitary effusion [15,41,42].

Finally, when once again combining the results of the two studies of the dogs with abdominal and pleural effusion reported in Table 2 [41,42], 48 out of the 103 dogs had plasma FDPs, D-dimer and fibrinogen concentrations, suggesting PF/PHF [12,29,66]. In addition, 23 dogs also showed signs of SF (defined by normal to increased FDPs and increased D-dimer concentrations with fibrinogen concentration > 100 mg/dL) or SHF (defined by normal to increased FDPs and increased D-dimer concentrations with fibrinogen concentration ≤ 100 mg/dL). The number of dogs with intracavitary effusion having an activated fibrinolytic system (i.e., PF/PHF or SF/SHF; *n* = 71) was significantly higher as compared with the dogs of the two control populations having an activated fibrinolytic system (i.e., the sick dogs without effusion [*n* = 39] and the clinically healthy control dogs [*n* = 9]; *p* < 0.00001; Table 3).

There are no other studies in veterinary medicine which correlate the systemic hyperfibrinolysis and the presence of intracavitary effusion with the exception of one study in which this association was made for dogs presenting with hemorrhagic ascites [55]. In human medicine, this association has also been found for patients with ascites due to liver diseases [75,76], and in patients receiving peritoneovenous shunting for intractable ascites [36,37,38,80].

## 7. Detection of Fibrinolysis in Dogs with Intracavitary Effusion Using ROTEM

Point-of-care viscoelastic hemostatic analyzers, such as ROTEM, can detect systemic changes, including fibrinolysis, in in vivo coagulation [81], although it cannot differentiate whether it is primary or secondary in nature. Fibrinolysis can be quantified using clot lysis parameters, such as the EXTEM lysis index at 60 min (LI60), the EXTEM maximum lysis (ML), and the EXTEM clot amplitude at 5 min (A5) [81]. In humans, FIBTEM seems to be even more sensitive than EXTEM in detecting fibrinolysis [81]. A study on 32 dogs with intracavitary effusion (25 with only abdominal effusion, six dogs with concomitant abdominal and pleural effusion, and one dog with only pleural effusion) aimed at comparing, using ROTEM, systemic fibrinolysis in these dogs and in 32 control dogs without intracavitary effusion individually matched to the dogs studied based on their type of underlying disease [15]. Four relevant results emerged from this study. First, EXTEM A5, EXTEM LI60 and EXTEM ML were significantly smaller, lower, and higher, respectively, in the 32 dogs with intracavitary effusion as compared with the control group of dogs without intracavitary effusion. This confirmed the presence of increased fibrinolysis in dogs with intracavitary effusion, even when studied using ROTEM [15]. Second, using a fibrinolytic severity score based on EXTEM LI60 and fibrinogen concentration [15], the dogs with intracavitary effusion showed a significantly lower degree of hypofibrinolysis (i.e., LI60 > the reference interval) and basal fibrinolysis (i.e., LI60 within the reference interval), and a higher degree of increased fibrinolysis (i.e., LI60 < the reference interval and plasma fibrinogen concentration > 100 mg/dL) and hyperfibrinolysis (i.e., LI60 < the reference interval and plasma fibrinogen concentration ≤ 100 mg/dL) as compared with dogs without intracavitary effusion. Third, the ROTEM lysis pattern, as classified by Schöchl et al. [20], also showed that dogs with intracavitary effusion had a significantly lower degree of hypofibrinolysis and basal fibrinolysis, and a higher degree of late, intermediate, and fulminant fibrinolysis as compared with dogs without intracavitary effusion. Fourth, while the fibrinolytic system was enhanced in the 32 dogs with intracavitary effusion as compared with the controls, both when measured using FDPs, D-dimer and fibrinogen concentrations, and when measured using ROTEM and fibrinogen concentrations, the two different methodologies had poor concordance in assessing the grade of severity of fibrinolysis in individual dogs [15]. The poor concordance of these two methods may have been due to the lack of sensitivity in detecting mild to moderate fibrinolysis using ROTEM [2,82,83] or, alternatively, to the fact that, while ROTEM assesses the plasmin activity in real time, the FDPs and the D-dimer concentrations measure the results of plasmin activity after its onset and beyond its duration of action [84].

Only a few other studies in veterinary medicine have investigated the presence of fibrinolysis in dogs with intracavitary effusion using viscoelastometry [46,50,51,55,85]. In two of these studies, dogs with intracavitary effusion constituted the minority of the population studied, namely 10 out of 47, and their coagulation characteristics were not specified, preventing any conclusions from being drawn regarding the degree of fibrinolysis present in these dogs [46,50]. In another study which investigated ROTEM parameters in 40 dogs with hemoperitoneum, no dogs showed hyperfibrinolysis [85]. On the opposite, hyperfibrinolysis was detected in all 28 dogs with spontaneous hemoperitoneum in a different study which used t-PA activated TEG [55]. Differences between the incidence and the severity of fibrinolysis, or differences in sensitivity in detecting fibrinolysis between ROTEM and t-PA activated TEG may account for these contrasting results [85].

## 8. Coagulation Status in Dogs with Ascites Due to Right-Sided Congestive Heart Failure

A systemic hypercoagulable state exists in many human, feline and canine patients with cardiomyopathy [86,87,88,89]. In humans, this hypercoagulable state may be associated with platelet activation, increased thrombin activity, endothelial dysfunction, and left atrial or ventricular enlargement [86,90], and may lead to thromboembolic events. Although still controversial, anticoagulant therapy is therefore sometimes started in human and feline patients with cardiomyopathy [91,92].

The major complication of anticoagulant therapy is the precipitation of bleeding events, and severe heart disease has been associated with an increased risk of bleeding during oral anticoagulant therapy [93]. Having shown that ascites has intrinsic fibrinolytic activity [72] and that this fibrinolytic activity is associated with PHF in dogs [42], another study investigated whether the presence of ascites due to right-sided congestive heart failure (CHF) was associated with PHF and bleeding in dogs [40]. To achieve this goal, the coagulation profiles of 20 ascitic dogs with right-sided CHF and 60 control dogs without intracavitary effusion (40 sick dogs without cardiac disease and 20 dogs with left-sided CHF) were studied. The hemostatic parameters analyzed were aPTT (activated partial thromboplastin time), PT (prothrombin time), and plasma concentrations of fibrinogen, FDPs, and D-dimer [40]. The occurrence of increased plasma FDPs and normal D-dimer concentrations with plasma fibrinogen concentrations > 100 mg/dL or equal/below this cut-off, suggestive of PF/PHF, respectively, was significantly more prevalent in dogs with cardiogenic ascites (18/20 [90%; PF = 12, PHF = 6]) as compared with the control dogs (5/40 [13%; all with PF] in the sick dogs without cardiac disease and 5/20 [25%; all with PF] in the dogs with left-sided CHF) [40]. The more prolonged aPTT and PT in dogs with cardiogenic ascites as compared with the control dogs additionally supported the presence of PF/PHF in these dogs [40]. Moreover, two dogs with right-sided CHF and PHF also had clinical evidence of prolonged bleeding after iatrogenic venipuncture, while signs of iatrogenic bleeding were not reported in any of the control dogs [40].

It is also interesting to note that, in a recent study, cats with cardiogenic pleural effusion had a reduced risk to of having aortic thromboembolism [45]. The findings of these studies in cats and dogs [40,45], combined with the known increased risk of bleeding during anticoagulant therapy in people with cardiac disease, should be an incentive for investigating whether PHF could also be a potential cause of bleeding in human patients with similar clinical presentations.

## 9. Possible Pathophysiology of Systemic Fibrinolysis in Dogs with Intracavitary Effusion

The results of the studies presented in these narrative reviews have demonstrated an association between intracavitary effusion and PF/PHF in dogs [15,40,41,42]. It was not possible in this narrative review to demonstrate the causal mechanisms between intracavitary effusion and the occurrence of PF/PHF in dogs. Nevertheless, despite the fact that abdominal and pleural fluids have traditionally been regarded by physicians to be inert fluids, they contain all of the proteins/enzymes present in plasma [70]. Thus, they include those which are involved in coagulation and fibrinolysis [75,94,95], but in an environment (i.e., the peritoneal and pleural cavities) in which their actions are no longer well regulated. The mesothelial cells lining the pericardium, pleural space and peritoneum play a fundamental role in creating this anticoagulant environment [96], primarily by means of the secretion of tPA and uPA which cleave the plasminogen found in the intracavitary fluids [97,98]. Mesothelial cells can additionally enhance anticoagulation by increasing the local expression of protein C [99]. This anticoagulant environment results in the formation of a fluid with inherently increased fibrinogen/fibrinolytic activity [70,75,76] in which there is a continuous exchange with the systemic circulation [75,79]. When the ascitic and pleural fluids re-enter the systemic circulation via the thoracic duct and the pulmonary veins, they might therefore contribute to the systemic hyperfibrinolytic state found in dogs with intracavitary effusion [36,37,38,39,75,76,80].

## 10. Conclusions

Under physiological conditions, fibrinolysis plays an important role in regulating the size of a clot [2], preventing the unnecessary accumulation of intravascular fibrin and enabling the removal of thrombi [5]. If plasmin activation exceeds the neutralizing capacity of the plasmin inhibitors, severe bleeding could potentially occur when there is PHF or SHF [6]. Similar to studies involving humans which have documented the association between intracavitary effusion and enhanced fibrinolysis [36,37,38,39,75,76,80], the results of the studies presented in this review led to the following conclusions:1.In nearly all the pleural and ascitic fluids of dogs, there is evidence of coagulation activation and fibrinolysis. This phenomenon occurs regardless of the underlying mechanism which leads to intracavitary effusion formation.2.There is a statistical association between an enhanced systemic fibrinolysis (mainly due to PF and PHF) and the presence of intracavitary effusion in dogs. Therefore, while in the intracavitary fluid of these dogs there is an activation of the coagulation and of the fibrinolytic process, at the level of the systemic circulation only the fibrinolytic process seems to be activated without a hemostatic trigger.3.FDPs, D-dimer and fibrinogen concentrations, and ROTEM and fibrinogen concentrations can both detect a state of enhanced fibrinolysis in dogs with intracavitary effusion; however, they may have poor concordance in assessing the grade of severity of fibrinolysis in individual dogs.


In the future, additional research is necessary to understand the molecular mechanisms which trigger coagulation and fibrinolysis in the fluids which accumulate in the pleural space and in the abdominal cavities, and to assess whether there is only a statistical association between intracavitary effusion and enhanced fibrinolysis or if there is a causal relationship.

## Figures and Tables

**Figure 1 animals-12-02487-f001:**
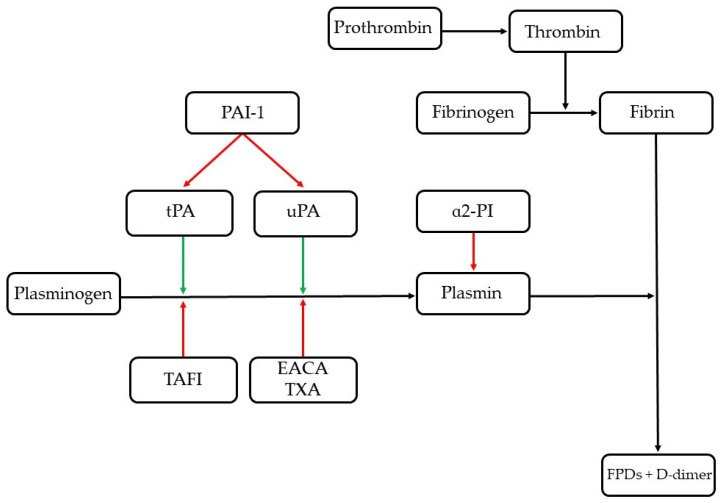
Fibrinolysis overview. The red arrows indicate inhibition; the green arrows indicate stimulation. α2-PI, α2-plasmin inhibitor; EACA, epsilon aminocaproic acid; FDPs, fibrin/fibrinogen degradation products; TAFI, thrombin-activatable fibrinolysis inhibitor; tPA, tissue plasminogen activator; TXA, tranexamic acid; uPA, urokinase plasminogen activator.

**Table 1 animals-12-02487-t001:** Principal components of the coagulation and fibrinolytic system.

	Synthesis Site	Major Role
Fibrinogen	Synthesized mainly by the liver	Fibrin precursor
Thrombin	Produced as prothrombin by the liver	Conversion of fibrinogen to fibrin
tPA	Synthesized and secreted by the endothelial cells	Conversion of plasminogen to plasmin
Plasminogen	Synthesized mainly by the liver	Major fibrinolytic protease after being converted to plasmin
uPA	Synthesized by monocytes, macrophages and the urinary epithelium	Conversion of plasminogen to plasmin
TAFI	Synthesized by the liver	Decreases binding of plasminogen to fibrin
α2-PI	Synthesized by the liver and platelets	Inhibits plasmin fibrilolytic activity
PAI-1	Synthesized mainly by the endothelial cells and hepatocytes	Inhibits tPA/uPA

α2-PI, α2-plasmin inhibitor; PAI-1, plasminogen activator inhibitor-1; TAFI, thrombin-activatable fibrinolysis inhibitor; tPA, tissue plasminogen activator; uPA, urokinase plasminogen activator.

**Table 2 animals-12-02487-t002:** Differences in the frequency of PF/PHF in dogs with intracavitary effusion as compared with sick dogs without effusion and clinically healthy control dogs (combined data from studies [41,42]). The sick dogs without effusion and clinically healthy dogs were individually matched for age, sex, and breed to the dogs with intracavitary effusion.

	PF/PHF	χ^2^-Test	*p*-Value	Comparison between Groups
G1. Dogs with effusion (*n* = 103)	48 (47%) (PF = 40; PHF = 8)			G1 vs. G2 χ^2^ = 8.34 *p* = 0.00388
G2. Sick dogs without effusion (*n* = 103)	28 (27%) (PF = 28; PHF = 0)	41.54	<0.00001	G1 vs. G3 χ^2^ = 41.69 *p* = 0.00001
G3. Clinically healthy dogs (*n* = 103)	7 (7%) (PF = 7; PHF = 0)			G2 vs. G3 χ^2^ = 15.18 *p* = 0.00009

G1, Group 1; G2, Group 2; G3, Group 3; PF, Primary fibrinolysis; PHF, Primary hyperfibrinolysis.

**Table 3 animals-12-02487-t003:** Differences in the frequency of PF/PHF, SF/SHF and the absence of fibrinolysis activation in dogs with intracavitary effusion as compared with sick dogs without effusion and clinically healthy control dogs (combined data from studies [41,42]). The sick dogs without effusion and the clinically healthy dogs were individually matched for age, sex, and breed to dogs with intracavitary effusion.

	PF/PHF	SF/SHF	Normal FDPs and D-Dimers	χ^2^-Test	*p*-Value	Comparison between Groups
G1. Dogs with effusion (*n* = 103)	48 (47%)	23 (22%)	32 (31%)			G1 vs. G2 χ^2^ = 20.16 *p* = 0.00004
G2. Sick dogs without effusion (*n* = 103)	28 (27%)	11 (11%)	64 (62%)	79.24	<0.00001	G1 vs. G3 χ^2^ = 41.69 *p* < 0.00001
G3. Clinically healthy dogs (*n* = 103)	7 (7%)	2 (2%)	94 (91%)			G2 vs. G3 χ^2^ = 24.52 *p* < 0.00001

FDPs, fibrin/fibrinogen degradation products; G1, Group 1; G2, Group 2; G3, Group 3; PF/PHF, primary fibrinolysis/primary hyperfibrinolysis; SF/SHF secondary fibrinolysis/secondary hyperfibrinolysis.

## Data Availability

Not applicable.

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
