# Peer review of "Fibrinolysis in Dogs with Intracavitary Effusion: A Review"

_animals, 2022, doi:10.3390/ani12192487_

Round 1

Reviewer 1 Report

Fibrinolysis is a complicated process, predicated on the conversion of inactive proenzymes to active forms, the ultimate goal of which is fibrin degradation. The central mediator of this process is plasmin. The primary substrate for physiological protein degradation by plasmin is the fibrin clot, keeping this process local. This manuscript reviewed fibrinolysis in dogs with intracavitary effusion. It is show that coagulation followed by fibrinogenolytic/fibrinolytic activity occur in any type of canine ascitic and pleural fluids. Moreover, systemic clotting abnormalities suggesting primary fibrinolysis/hyperfibrinolysis  occurs in dogs with intracavitary effusion. Enhanced fibrinolytic activity in dogs with intracavitary effusions can be also detected by rotational thromboelastometry despite the degree of agreement between ROTEM and FDPs, D-dimer. This manuscript is very well.

Reviewer 2 Report

Dear Authors,

The paper entitled “Fibrinolysis in dogs with intracavitary effusion: a review" submitted for publication in Animals concerns review of pathological states of fibrinolysis in dogs.

As authors wrote “The first aim of this review was to describe pathophysiology, classification and detection of fibrinolysis in clinical practice. The second aim was to summarize the finding in  the literature that refers to the intrinsic fibrinolytic activity of the intracavitary fluid in  dogs and to describe their systemic fibrinolytic status”. This systematic review is based on authors original papers in clinical aspect of pathologies of canine fibrinolysis. 

The paper represents high scientific work, and it can be accepted for publication in Animals after minor revision.

Major Comments:

Paragraph 2- the figure illustrated physiology and/or pathophysiology of fibrinolysis will be more reader friendly.

Paragraph 3-Classification of Fibrinolysis- I suggest change the sequence of: types of fibrinolysis: basal fibrynolisis, physiological fibrinolysis, primary fibrino(geno)lysis, and reactive or secondary fibrinolysis.

Line 127- the value of fibrinogen concertation should be compare with references values. However, the reference values depend on used laboratory technique which vary between different laboratories. Thus, I recommend to rather use fold increase/decrease in whole article instead of number values.

Line 313 and 337- table 1 and table 2. The table title includes the comparison of age, sex and breed in exanimated dogs, however  for me it is not clear where these comparisons are. Please clarify.

Paragraph 4-Please qualify which tests were performed in dogs and humans.

Paragraph 10. Conclusions- please delate citations in this part of the article; in this part of article we only conclude our work

Minor Comments:

Abstract:

Line 34: FDPs – please expand abbreviation

Line 35: PF/PHF – please expand abbreviation

Line 95- I suggest in this paragraph summarizing table of  key proteins involved in fibrinolysis could be added

Line 112 disseminated intravascular coagulation – lack of abbreviation

Line 147 fibrin-fibrinogen change into fibrin/fibrinogen

Line 230 – surface- lack of full stop

Line 271 -the end of the sentence- lack of full stop

Reviewer 3 Report

The manuscript contains:

ü  long sentences, broken by parentheses with lots of additional information in the brackets,  e.g. lines: 68 - 71. The reader quickly loses the context of the sentence - focusing attention on the content and understanding is very difficult.

ü  repeated sentences,  e.g. lines:  46 - 50 and 63 - 67,   50 - 51 and 97.

ü  not unified adjectives,  e.g. hemostatic (more commonly used) and haemostatic.

The authors

1. suggest the existence of multiple forms of antiplasmin (in the manuscript the    plural is used) - why ?

2.   suggest the existence of several mechanisms of fibrinolysis. Even if the etiology, the mechanism of fibrinolysis activation (plasminogen activation by tPA and/or contact factors) as well as the rate of it progression are different, the process of fibrynogen, unstabilized and stabilized fibrin degradation, is the same. Fibrinolysis is activated in response to the activation of the coagulation and is closely related to it (the processes should not be separated).

3.   present the division of fibrinolysis (lines 101 - 134) into different types, which is neither widely used (bears the signs of historical obsolescence) nor is currently recommended by the ISTH. The literature data on which the authors relied also does not suggest the existence of such a division and such interpretation.

4. present methods for detecting and observing fibrinolysis - however, their descriptions are very sparse and strongly unclear.

5.   describe hemostatic parameters in healthy dogs and in those with/without fluid accumulation in body cavities, as well as the fibrinolytic activity of these fluids (subsections 5, 6, 7). These descriptions are chaotic, strongly distracting to the reader (e.g., lines 319 - 327). They also do not correlate with the content of the tables to which they refer. In addition, the titles of the tables do not correlate with their content and there is no description of the symbols used therein.

Age, sex, breed, and the type of disease have a great influence on the parameters of hemostasis - this is mentioned in the titles of the tables but no longer in their contents. 

6. described hemostatic parameters in dogs with ascites caused by right-sided congestive heart failure in an inexplicit manner.

If in subsections 5 - 8:

- the sentences were shorter,

- the descriptions for each experimental group were separated,

- the characteristics of the dogs were approximated,

the manuscript would have been more readable and understandable.

7.   subsection 8 address dogs but the conclusions therein address humans.

8.  should not formulate general conclusions in relation to observations in humans. 
